

# Tuck-KGC: based on tensor decomposition for diabetes knowledge graph completion model integrating Chinese and Western medicine

Jiangtao ZhangSun[1], Yu Xin Yang[1], Beiji Zou[2], Qinghua Peng[1] and Xiao Xia Xiao[1]

[1] School of Informatics, Hunan University of Chinese Medicine, Changsha, Hunan, China
[2] School of Computer Science and Engineering, Center South University, Changsha, Hunan, China

Corresponding author
Xiao Xia Xiao,
amily_x@hnucm.edu.cn

## ABSTRACT

The medical knowledge graph is essential for intelligent medical services, encompassing personalized diagnostics, precision therapies, and intelligent consultations, among others. However, medical knowledge graphs frequently suffer from incompleteness, primarily due to the absence of certain entities or relationships. The incomplete nature of knowledge graphs poses substantial challenges to these tasks. Knowledge graph completion technology is instrumental in addressing this issue. Specifically, tensor decomposition-based approaches for knowledge graph completion embed entities and relationships into the vector space, where tensor decomposition computations are employed to predict missing relationships within the knowledge graph. However, the tensor representation of entities and their relationships often overlooks crucial entity type information, potentially resulting in an abundance of irrational relationships during the prediction process. To mitigate this, we propose the Tucker Decomposition Knowledge Graph Completion (Tuck-KGC) method, which incorporates entity types into the tensor decomposition framework. This method maps the types of medical entities to vectors, which are seamlessly integrated into the knowledge graph representation learning process. This allows the model to thoroughly absorb entity information, thereby enhancing the accuracy of link prediction. To evaluate the Tuck-KGC, we built the Dia dataset, a knowledge graph tailored for precision medical analysis, which integrates both Traditional Chinese Medicine and Western medicine perspectives. The Dia dataset encompasses 10,294 entities with 214 relationships, covering a comprehensive spectrum including diseases, treatments, clinical manifestations, complications, etiology, and so on. Building upon the Dia dataset, experimental results indicate that the Tuck-KGC model boosts link prediction accuracy by roughly 8%, affirming the efficacy of incorporating entity type information into the model.

## INTRODUCTION

Knowledge graphs (KGs), also known as semantic networks, represent a network of real-world entities such as objects, events, situations or concepts and illustrates the relationships between them. These entities and their relationships are often stored in a graph database using resource description framework (RDF)-style triples $(e_s, w_r, e_o)$, where $e_s$ and $e_o$ represent subject and object entities, and $w_r$ represents a relation that connects $e_s$ to $e_o$. KGs have been applied to many tasks, including link prediction, which can make intelligent-aided diagnosis, and treatment possible in the field of medicine. This can assist doctors in making decisions more efficiently. Moreover, the visualized knowledge graphs can also allow patients to understand diseases more clearly and easily. However, KGs are often incomplete, primarily due to the absence of certain entities or relationships. To address these challenges, many researchers aim to improve the accuracy and reliability of KGs by predicting the existence of relationships, a process often referred to as knowledge graph completion (KGC) (*Trouillon et al., 2017*). Advances in vector embeddings have led to the development of many embedding-based KGC algorithms including tensor decomposition-based approaches (*Dettmers et al., 2018*; *Trouillon et al., 2016*).

However, existing knowledge graph embeddings treat all entities as the same type, ignoring their assigned types. In knowledge graphs related to healthcare, particularly in the context of diabetes diagnosis and treatment by integrating Western and Chinese medicines, entities frequently undertake various roles in interconnections. For example, in the ternary group comprised of hypoglycemia, ADE_Disease, and renal failure the hypoglycemia entity type is a symptom of the disease, but the ternary group comprised of fasting, Reason_Disease, and hypoglycemia the hypoglycemia entity type is the disease. Hence, a clear demarcation is warranted among these types, particularly when knowledge-based reasoning comes into play. To enhance the accuracy and reliability of medical knowledge graphs derived from real clinical information, we proposed the Tucker Decomposition Knowledge Graph Completion (Tuck-KGC) method based on the TuckER (*Balažević, Allen & Hospedales, 2019*) model with fused entity type embedding. Additionally, we constructed a diabetes knowledge graph that integrates both traditional Chinese medicine and Western medicine to evaluate our methods.

Diabetes, a prevalent chronic condition, has emerged as a significant challenge within the global public health domain. Data released by the International Diabetes Federation indicates that the global prevalence of diabetes among adults exceeds 465 million, with projections indicating a steady rise in the coming years (*International Diabetes Federation, International Society of Nephrology, 2023*). To mitigate the risk of severe complications, individuals with diabetes require consistent blood glucose monitoring and pharmacological intervention. The construction of a comprehensive diabetes knowledge graph that integrates both traditional Chinese and Western medicine empowers patients by facilitating information access (*Xiong, Power & Callan, 2017*). Moreover, this knowledge graph can also support intelligent-question-answering (*Hao et al., 2017*) and intelligent-aided diagnostics (*Zhang et al., 2016*), thereby extending the reach of intelligent medical services beyond geographical and temporal limitations. Compared to the prolonged drug

dependence resulting from Western medicine's pharmacological control and the challenges faced by Chinese medicine in providing immediate relief for diabetes, the synergy of both Chinese and Western medicine offers a more holistic and multifaceted treatment approach.

Incorporating the sophisticated fusion of Chinese and Western medicine practices, China has developed a unique and potent healthcare system that significantly strengthens clinical research initiatives. These endeavors are pivotal in enhancing the capabilities of an integrated approach to medicine in the prevention and treatment of major chronic diseases (*Dobos & Tao, 2011*). To address the intricate challenges associated with people's health and diseases, the integration of Chinese and Western medicine in the diagnosis and treatment of diabetes mellitus merges the Traditional Chinese Medicine practices of tongue and pulse diagnosis with the Western medicine advancements in biochemical imaging and other diagnostic techniques. This holistic strategy merges the comprehensive diagnostic insights of Chinese medicine with the targeted therapeutic methods of Western medicine. By leveraging the strengths of both Chinese and Western medical traditions (*Wang & Zhang, 2017*; *Zhang et al., 2010*), it endeavors to offer more precise diagnosis and treatment services for those afflicted with diabetes. This method analyzes the disease from diverse medical perspectives, pinpointing the cause and local pathological changes, while considering the overall response and the dynamic shifts that occur throughout the disease's progression (*Zhang et al., 2010*; *Wang & Zhang, 2017*). Through the harmonious integration and complementation of both medicines, the provision of personalized patient care is not only feasible but also serves to enhance therapeutic effectiveness (*Yang, 2010*).

To delve into intelligent diagnostic and therapeutic methodologies for diabetes, we built a diabetes knowledge graph utilizing the DiaKG (*Chang et al., 2021*) dataset from Ruijin Hospital on the Ali Tianchi platform, diabetes data extracted from Traditional Chinese Medicine diagnostic criteria, and SDKG-11 (*Zhu et al., 2022*) dataset. This KG harmoniously merges insights from both Chinese and Western medicine. We further enhance this graph by integrating entity type embedding using TuckER (*Balažević, Allen & Hospedales, 2019*). Although our constructed knowledge graph of diabetes contains knowledge derived from multiple databases, it is still incomplete. To address the issue of incomplete medical knowledge graphs, we introduce the Tuck-KGC, tailored to infer missing connections between medical entities. This study conducts a comparative empirical analysis between the prevailing baseline model and the approach introduced herein. The comparison is grounded on a knowledge graph we have built, which encapsulates the diabetes-related knowledge from both Chinese and Western medicine. The findings indicate that the method proposed in this paper is not only advanced but also significantly outperforms the mainstream baseline model.

## RELATED WORK

In recent years, extensive research has been dedicated to the application of knowledge graph embedding (KGE) techniques for addressing the challenge of knowledge graph completion. SimplE (*Kazemi & Poole, 2018*), a linear model based on the Canonical Polyadic (CP) decomposition (*Hitchcock, 1927*) from 1927, stands out for its approach. It learns two

distinct embedding vectors for each entity that is linked within the knowledge graph. However, CP decomposition typically underperforms in link prediction tasks. The SimplE model, a straightforward adaptation of CP decomposition, enables the learning of two separate embedding vectors for every entity. The complexity of SimplE increases linearly with the size of the embedding. DistMult (*Yang et al., 2014*) presents a comprehensive neural network framework for multi-relational representation learning, characterized by a diagonal matrix for each relation. This design effectively captures interactions within relational data while mitigating overfitting. However, the linear transformation applied to entity embedding vectors in DistMult imposes certain limitations. The binary tensor learned by DistMult displays symmetry across both the subject and object entity modes, which poses a challenge in accurately representing asymmetric relationships.

The TuckER (*Balažević, Allen & Hospedales, 2019*) model, grounded in the Tucker decomposition (*Tucker, 1966*) principle, reprints a seminal, pioneering approach in the realm of third-order tensor decomposition for learning knowledge graph representations. This approach adeptly dissects a tensor into a collection of matrices and a more compact core tensor. In addition, TuckER (*Balažević, Allen & Hospedales, 2019*) shows that tensor decomposition-based embedding models are capable of effectively representing knowledge graphs, as there exist embeddings for entities and relations that can successfully differentiate between true and false triples for every piece of information involving all entities and relations.

The RotatE (*Sun et al., 2019*) model advances this approach by mapping entities and relations into the complex vector space, representing each relation as a rotation connecting the source entity to the target entity. This approach has been shown to be highly effective in modeling three fundamental relationship types: symmetry/antisymmetry, inversion, and composition. Furthermore, the linear scalability of RotatE (*Sun et al., 2019*) in both time and memory makes it suitable for application in extensive knowledge graphs.

Table 1 shows a summary of link prediction models. Each link prediction model has a scoring function and each function has its own dimension of relation parameters. These models also possess significant terms in their space complexity: $d_e$ and $d_r$ are the dimensionalities of entity and relation embeddings, while $n_e$ and $n_r$ denote the number of entities and relations, respectively, $e_s, W_r \in \mathbb{R}^{d_\omega \times d_h}$ denote a 2D reshaping of $e_s$ and $W_r$, respectively, $h_{e_s}$ and $t_{e_o} \in \mathbb{R}^{d_e}$ are the head and tail entity embedding of entity $e_s$, and $W_{r^{-1}} \in \mathbb{R}^{d_r}$ is the embedding of relation $r^{-1}$ (which is the inverse of relation $r$), $*$ is the convolution operator, $\langle \cdot \rangle$ denotes the dot product and $\times_n$ denotes the tensor product along the n-th mode, $f$ is a non-linear function, and $\mathfrak{W} \in \mathbb{R}^{d_e \times d_r \times d_e}$ is the core tensor of a Tucker decomposition.

ConvE (*Dettmers et al., 2018*) tackles the knowledge graph completion task using of a neural network model. It accomplishes this by employing a global 2D convolution operation on the embedding vectors corresponding to the head entity and relationship. These vectors are initially converted into matrices and then concatenated. The derived feature layer is unwrapped, processed through a linear layer transformation, and subsequently used to calculate a score for each ternary relationship by computing inner products with all object entity vectors.

**Table 1  A summary of link prediction models.**

| Model | Scoring function | Relation parameters | Space complexity |
|---|---|---|---|
| SimplE (*Kazemi & Poole, 2018*) | $\frac{1}{2}\left(\langle h_{e_s}, W_r, t_{e_o}\rangle + \langle h_{e_o}, W_{r^{-1}}, t_{e_s}\rangle\right)$ | $W_r \in \mathbb{C}^{d_e}$ | $\mathfrak{O}(n_e d_e + n_r d_e)$ |
| DistMult (*Yang et al., 2014*) | $\langle e_s, W_r, e_o\rangle$ | $W_r \in \mathbb{R}^{d_e}$ | $\mathfrak{O}(n_e d_e + n_r d_e)$ |
| RotatE (*Sun et al., 2019*) | $-\|h \circ r - t\|^2$ | $W_r \in \mathbb{C}^{d_e}$ | $\mathfrak{O}(n_e d_e + n_r d_e)$ |
| ConvE (*Dettmers et al., 2018*) | $f\left(\text{vec}\left(f\left([e_s; W_r] * \omega\right)\right) W\right) e_o$ | $W_r \in \mathbb{R}^{d_r}$ | $\mathfrak{O}(n_e d_e + n_r d_r)$ |
| TuckER (*Balažević, Allen & Hospedales, 2019*) | $\mathfrak{W} \times_1 e_s \times_2 w_r \times_3 e_o$ | $W_r \in \mathbb{R}^{d_r}$ | $\mathfrak{O}(n_e d_e + n_r d_r)$ |
| Tuck-KGC (ours) | $((\mathfrak{W} \times_1 e_s \times_1 e_\tau) \times_2 w_r) \times_3 e_o \times_3 e_\tau$ | $W_r \in \mathbb{R}^{d_r}$ | $\mathfrak{O}(n_e d_e + n_r d_r)$ |

Both linear models and neural network models designed for knowledge graph completion fail to encapsulate entity type information, a crucial aspect in real-world applications. To address this, we integrate entity types into the knowledge graph completion model, harnessing their effectiveness to enhance our methodology. TuckER (*Balažević, Allen & Hospedales, 2019*), founded on tensor decomposition for knowledge graph information representation, further maps entity types into a vector space, facilitating the construction of the entity type matrix within the learning process of the model. Empirical experiments corroborate that integrating entity types significantly enhances model accuracy.

## METHODS

### Knowledge embedding

To effectively incorporate entity type information and knowledge graph triples into the vector space, we employ the TransE embedding method. The ultimate embedding of an entity is generated through the interaction of its associated entity category information auxiliary representation. For head entity embedding, we multiply the vector of head entity representation with the auxiliary entity type vector to obtain the final representation of the head entity. Similarly, we adopt the same approach to get the final representation of the tail entity. The equations are as follows:

$$\|e_s \circ e_\tau + w_r - e_o \circ e_\tau\|. \tag{1}$$

Where $e_s, e_o \in \mathbb{R}^{d_e}$ represent the embedding vectors of the head entity and tail entity, $e_\tau$ represent the entity type embedding vectors, $w_r \in \mathbb{R}^{d_r}$ represent relation embedding vectors.

### Link prediction

Given a knowledge graph $G = (E, R, T)$, where $E$ and $R$ represent the sets of entities and relationships, respectively, and $T$ represents the set of triples in the format of $(e_s, w_r, e_o) \subset E \times R \times E$; Tuck-KGC is an approach to solving the knowledge graph completion problem through the link prediction task, which aims to construct new triples using the existing set of triples $T$ and entity types $E_\tau$. Tuck-KGC addresses the issue of knowledge graph incompleteness by engaging in the link prediction task, with the objective of constructing a new triple $(e'_s, w'_r, e'_o)$, where $e'_s, e'_o \in E, w'_r \in R$. On this basis, a

suitable scoring function $\varphi$ is used to determine the reasonableness of the triad and its plausibility and to predict the new relationship, where the scoring function is defined as $\varphi(e_s, w_r, e_o) : E \times R \times E \longrightarrow R$. The new triple $(e'_s, w'_r, e'_o)$ is generated by replacing one of the items that belongs to the existing set of triples $(e_s, w_r, e_o)$, where the item belongs to the same category as the replacement item. The freshly created triples are subsequently assessed through the scoring function $\varphi$, thereby converting the link prediction task into a sorting challenge. In this process, the scoring function $\varphi$ evaluates each of the newly generated triples, followed by their arrangement in descending order of score.

## Tuck-KGC implementation

To ensure the triad's bidirectional prediction, we utilize the same $\mathfrak{E}$ embedding matrix to portray the head and tail mapping modelling $A, C$ matrices, respectively, *i.e.*, $\mathfrak{E} = A = C \in \mathbb{R}^{n_e \times d_e}$. At the same time, we map the relational embedding matrix R to the B matrix in the tensor decomposition modelling, *i.e.*, $\mathfrak{R} = B \in \mathbb{R}^{n_r \times d_r}$. Upon this foundation, we incorporate an entity type matrix denoted as $\Gamma \in \mathbb{R}^{n_\tau \times d_\tau}$ which we use to refine the entity matrix operation. This integration of entity type information enriches the semantic representation of entities in the link prediction task. Additionally, technical abbreviations will be explained upon initial use.

In this context, $n_e$ and $n_r$ represent the respective quantities of entities and relations, while $d_e, d_r, d_\tau$ signify the dimensions of the embedding vectors for entities, relations, and entity types. The scoring function for the Tuck-KGC model, which is based on the tensor decomposition of fused-type embeddings, is formulated as follows:

$$\varphi(e_s, w_r, e_o, e_\tau) = \left( \left( \mathfrak{W} \times_1 e_s \times_1 e_{s_\tau} \right) \times_2 w_r \right) \times_3 e_o \times_3 e_{o_\tau} \tag{2}$$

where $\mathfrak{W} \in \mathbb{R}^{d_e \times d_r \times d_e}$ represents the core tensor, $e_s, e_o \in \mathbb{R}^{d_e}$ are the rows of $\mathfrak{E}$ representing the embedding vectors for the head entity and tail entity, $e_{s_\tau} or e_{o_\tau}$ the rows of $\Gamma$ representing the head entity type or tail entity type embedding vectors, $w_r \in \mathbb{R}^{d_r}$ the rows of $R$ representing relation embedding vectors, $\times_n$ denotes the tensor product along the n-th mode. Equation (2) describes the methodology by which Tuck-KGC prioritizes all triples in the dataset, integrating the entities and their corresponding relationships with their specific entity types. The triple with the highest ranking is subsequently designated as the prediction outcome.

Throughout the training phase, we map triples and entity types into a vector space and input the resulting vectors into the model. We then score each triple and calculate the loss value through the model's scoring function. Parameters are updated using backpropagation. The model undergoes repeated training iterations until the loss value is minimized, and the best results are achieved. The loss function for an entity-relationship pair in comparison to all other entities is defined as:

$$\text{Loss} = -\frac{1}{n_e} \sum_{i=1}^{n_e} \left( y^{(i)} \log\left(p^{(i)}\right) + \left(1 - y^i\right) \log\left(1 - p^{(i)}\right) \right). \tag{3}$$

Where $n_e$ represents the dimension of the entity embedding vector space, $p^{(i)}$ signifies the vector of predicted probabilities, and $y^{(i)}$ is the binary label vector. To combat overfitting,

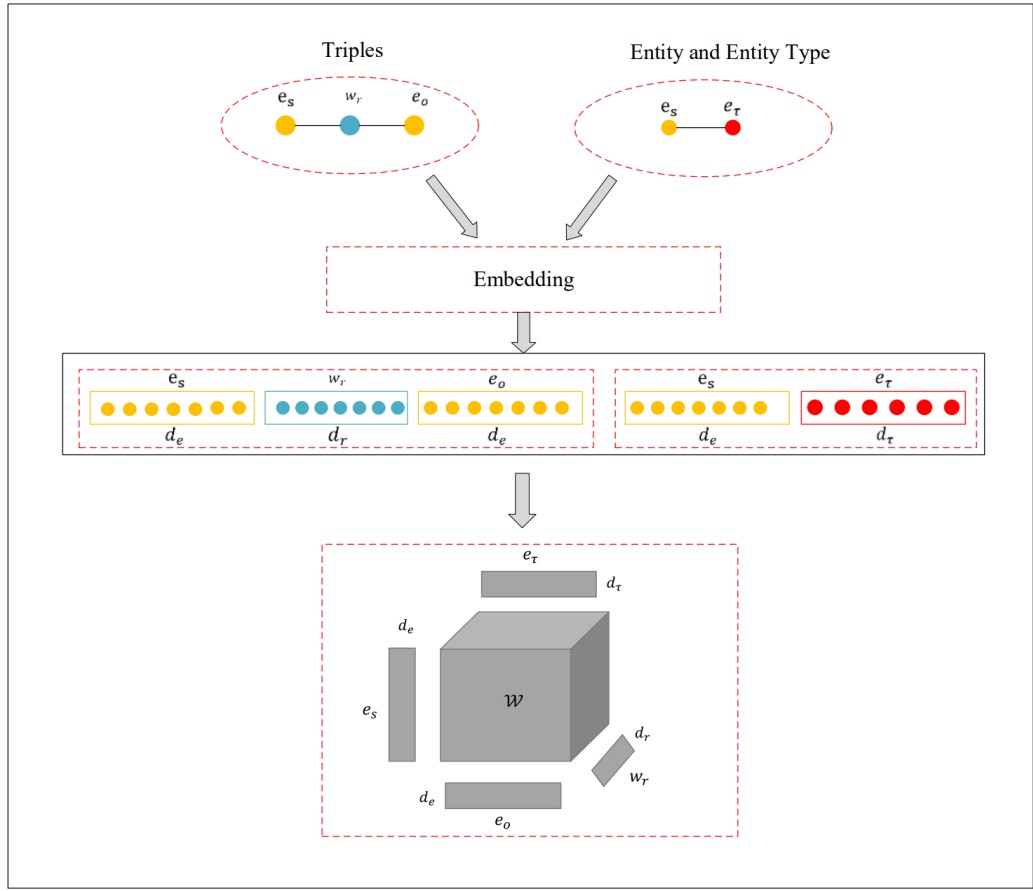

**Figure 1** **Visualization of the Tuck-KGC architecture.**

this research incorporates dropout, complemented by the Adam optimizer and label smoothing techniques during the training process. The detailed architecture of the model can be observed in Fig. 1.

## Experiments
## Datasets

The Dia dataset employed in this investigation comprehensively includes data from both Traditional Chinese medicine (TCM) and Western medicine diabetes data. The Western medicine dataset was sourced from the DiaKG (*Chang et al., 2021*) dataset available from Ali Tianchi Ruijin Hospital and from the diabetes-related section of SDKG-11 established by *Zhu et al. (2022)*. The DiaKG dataset includes 22,050 medical entities and 6,890 annotated entity relationships, incorporating insights from 41 diabetes experts. The dataset covers various types, including medication usage, clinical cases, diagnosis and treatment methods, and clinical research. The SDKG-11 dataset, created by *Zhu et al. (2022)*, comprises triads extracted from publications indexed in PubMed from the year 2020, which boast an impact factor of 2.0 or higher. We meticulously isolated the diabetes-related subset, which primarily consists of fundamental research and theoretical foundations of diabetes.

The Traditional Chinese Medicine diabetes dataset, used in this study is derived from the TCM guidelines for diabetes and its related complications (*Diabetes Branch of the Chinese Association of Traditional Chinese Medicine, 2011a*; *Diabetes Branch of the Chinese Association of Traditional Chinese Medicine, 2011b*). The TCM diabetes data subset also incorporates *Li & Zhang (2023)* perspective of syndrome differentiation based on syndrome elements, which is constructed manually following a main-predicate-object structure (*Fang et al., 2017*; *Yang, 2022*). Firstly, medical terminology entities, their attributes, and relationships between these entities are extracted from his monographs by human labor. Finally, based on TCM theories, the perspectives of *Zhu (2006)*, and entity relationships, triples are constructed, which are ultimately merged into the TCM diabetes data subset. The knowledge graph that incorporates *Zhu (2006)* thoughts on syndrome differentiation and treatment can better support the construction of intelligent-aided diagnosis models based on traditional Chinese medicine theory.

Due to the presence of biomedical entities synonyms and multiple layers of semantic duplication, duplicate triples can be found in the Diabetes dataset for both Chinese and Western medicine. Initially, the MTransE (*Chen et al., 2016*) model was used to align entities in the raw data. Based on *Gong et al. (2021)* method of integrating Chinese and Western medicine knowledge to construct a diabetes knowledge graph, we also incorporated entity type information and developed a concept definition module, entity classification module, an attribute division module, and relationship matching module.

Initially, we input Chinese and Western medicine entities into the entity concept layer and established the disease layer, clinical performance layer, and treatment layer. Subsequently, we used the entity classification module to input various generic entities into the entity concept layer, classifying them based on diabetes symptoms and the corresponding treatments and protective measures. The classification process is to classify various common entities related to traditional Chinese and Western medicine according to their relevance to diabetes symptoms and their treatment and prevention methods (*Wei, 2021*). Subsequently, we input the types of diabetes symptoms along with the corresponding TCM treatments and prescriptions as documented in the TCM treatment process. We then provide the clinical manifestations of diabetes, its complications, and the corresponding treatments and protective methods in Western medicine. Additionally, we list the distinct entities associated with TCM and Western medicine separately. The attribute classification module categorizes the TCM and Western medicine entities in the treatment layer according to their attributes. The relationship matching module, was employed to identify the potential relationships between Chinese and Western medicine entities, thereby providing a theoretical foundation for the interconnection between these entities and the generic entities. Finally, the construction module connects the entities based on the entity's inherent relationship in the entity concept layer and the types determined by the entity classification module, resulting in the formation of a ternary-based knowledge graph. This process yielded a total of 10,294 medical entities and 12,863 pairs of entity relationships, as shown in Table 2. For a detailed breakdown of the entity concept layer, please refer to Appendix S1 (Tables SA1–SA3).

**Table 2  Dia dataset statistics.**

|  | Dia | Train | Value | Test |
|---|---|---|---|---|
| Entities | 10,294 | 5,334 | 1,247 | 1,369 |
| Relations | 214 | 214 | 214 | 214 |
| Triples | 12,863 | 7,717 | 2,573 | 2,573 |

The construction of the Dia dataset, integrating diagnostic and therapeutic knowledge of diabetes from both Traditional Chinese medicine and Western medicine, will be divided at random into training, validation, and test subsets in this research with proportion of 0.6, 0.2, and 0.2, respectively.

## Evaluation indicators

In this paper, we reference the evaluation metrics introduced by *Balažević, Allen & Hospedales (2019)* to demonstrate the efficacy of our model in the link prediction task.

We utilize standard evaluation metrics commonly employed in the link prediction literature, namely, mean reciprocal rank (MRR) and Hits@k (where $k = 1, 3, 10$). The mean reciprocal rank is computed as the average of the reciprocals of the mean ranks assigned to the correct triple among all candidate triples:

$$\text{MRR} = \frac{1}{|T_{\text{test}}|} \sum_{(h,r,t) \in T_{\text{test}}} \frac{1}{2} \left( \frac{1}{\text{rank}_{r,t}(h)} + \frac{1}{\text{rank}_{h,r}(t)} \right) \qquad (4)$$

where $\text{rank}_{r,t}(h)$ denotes head entity ordering, and $\text{rank}_{h,r}(t)$ denotes tail entity ordering. The value of $MRR \in (0, 1)$ ranges from MRR and the larger the calculated value, the better the performance of the model link prediction. Hits@k quantifies the percentage of instances where a true triple is positioned among the top $k$ candidate triples:

$$\text{Hits@}k = \frac{1}{|T_{\text{test}}|} \sum_{(h,r,t) \in T_{\text{test}}} \frac{1}{2} \left( \left| \{ (h,r,t) | \text{rank}_{r,t}(h) \leq k \} \right| + \left| \{ (h,r,t) | \text{rank}_{r,t}(t) \leq k \} \right| \right). \qquad (5)$$

Hits@k focuses on the overall ranking, with larger values indicating that the model performs better in the link prediction task.

## Baseline and parameter

We compare Tuck-KGC with other link prediction models, including TransE (*Bordes et al., 2013*), DistMult (*Yang et al., 2014*), ComplEx (*Trouillon et al., 2016*), SimplE (*Kazemi & Poole, 2018*), relational graph convolutional neural networks (R-GCN) (*Schlichtkrull et al., 2018*), ConvE (*Dettmers et al., 2018*), InteractE (*Vashishth et al., 2020*), and RotatE (*Sun et al., 2019*), where entities and relations are embedded into vector space through linear operations; DistMult (*Yang et al., 2014*), is a general neural network framework for multi-relational representation learning; ComplEx (*Trouillon et al., 2016*) extends DistMult to the complex domain; SimplE (*Kazemi & Poole, 2018*) which encodes each entity with two separate embedding vectors entity actual connection in the knowledge graph; R-GCN (*Schlichtkrull et al., 2018*) is an extension of graph convolutional networks for relational data; ConvE (*Dettmers et al., 2018*), which utilizes 2D convolution to learn

**Table 3  The detailed parameter list of the Tuck-KGC model.**

| Parameter | Value |
| --- | --- |
| Entity Embedding Dimension | 400 |
| Relationship Embedding Dimension | 400 |
| Entity Type Embedded Dimensions | 400 |
| Batch size | 256 |
| Learning rate | 0.0003 |
| Label smoothing | 0.1 |

deep features of entities and relations; InteractE (*Vashishth et al., 2020*) replaces the simple feature reshaping used in ConvE with check reshaping and circular convolution; RotatE (*Sun et al., 2019*) charactering each relation as a rotation from a source entity to a target entity in a complex vector space.

The Tuck-KGC model is implemented in Pytorch unifying the Tuck-KGC and baseline models on NVIDIA Gefiorce RTX3090Ti GPU. We establish the embedding dimension for both entities and relations as $d_e = d_r = 400$, the embedding dimension of entity types to $d_\tau = 400$, the learning rate selected from the set {0.01,0.005,0.003,0.001,0.0005,0.0003}, and the learning decay rate to be chosen from {1.0,0.998,0.995,0.99}. Experimental outcomes reveal optimal performance at a learning rate and learning rate decay of (0.0003,1.0). Furthermore, our experiments demonstrate the dropout value of (0.3,0.4,0.4,0.5) can effectively mitigates overfitting. The baseline models mentioned are implemented using OpenKE (*Han et al., 2018*) toolkit. Table 3 shows the detailed parameter list of the Tuck-KGC model.

## RESULTS

The results of all link prediction task results are presented in Table 4 (where the top-performing and second results are distinguished by boldface and underscoring, respectively and the RotatE (*Sun et al., 2019*) results are presented without the self-adversarial negative sampling for an equitable comparison). As shown in Table 4, our enhanced Tuck-KGC model outperforms the TuckER (*Balažević, Allen & Hospedales, 2019*) model across all four evaluation metrics, including a 10% increase in Hits@10 and an 8% increase in the remaining metrics. This demonstrates the efficacy of introducing entity category vectors in our tensor decomposition approach to enhance model performance. Compared to conventional linear models such as TransE (*Bordes et al., 2013*), DistMult (*Yang et al., 2014*), and SimplE (*Kazemi & Poole, 2018*), Tuck-KGC possesses a considerable advantage. Tuck-KGC effectively captures the intricate interactions between entities and relations using its core tensor, allowing for better modeling of complex relationships. Additionally, Tuck-KGC reduces the model's parameters by representing high-dimensional tensors with a set of low-dimensional vectors and matrices. While ComplEx (*Trouillon et al., 2016*) is a linear model that gains from multi-task learning, Tuck-KGC outperforms it by about 15% in the most stringent evaluation metric Hits@1.

**Table 4   Evaluation results on Dia.**  Best results are in bold and second best results are underlined.

| | Dia | | | |
| --- | --- | --- | --- | --- |
| | MRR | Hits@10 | Hits@3 | Hits@1 |
| TransE (*Bordes et al., 2013*) | .241 | .341 | .276 | .198 |
| DistMult (*Yang et al., 2014*) | .323 | .446 | .301 | .199 |
| ComplEx (*Trouillon et al., 2016*) | .337 | .424 | .387 | .351 |
| SimplE (*Kazemi & Poole, 2018*) | .441 | .427 | .399 | .368 |
| R-GCN (*Schlichtkrull et al., 2018*) | .255 | .433 | .289 | .158 |
| ConvE (*Dettmers et al., 2018*) | .443 | .522 | .463 | .411 |
| InteractE (*Vashishth et al., 2020*) | .501 | .509 | .506 | .419 |
| RotatE (*Sun et al., 2019*) | **.583** | .576 | .522 | .425 |
| TuckER (*Balažević, Allen & Hospedales, 2019*) | .452 | .566 | .488 | .433 |
| Tuck-KGC (ours) | .561 | **.667** | **.575** | **.497** |

When contrasted with sophisticated deep learning neural network models, such as R-GCN (*Schlichtkrull et al., 2018*), ConvE (*Dettmers et al., 2018*), and InteractE (*Vashishth et al., 2020*), Tuck-KGC surpasses deep learning neural network models due to its simpler configuration and a limited number of parameters. Moreover, Tuck-KGC performs marginally worse than RotatE (*Sun et al., 2019*), with a difference of approximately 2% in the MRR evaluation metric. Our analysis concludes that RotatE (*Sun et al., 2019*) employs a distinctive self-adversarial negative sampling technique, defining each relationship as a rotation from a source entity to a target entity within a complex vector space. This capability enables the model to capture and infer a broad spectrum of relational patterns, including symmetric, asymmetric, inversion, and combination relationships.

## Influence of parameter

To test the performance of the Tuck-KGC model, we conducted a comparative analysis focusing on the entity embedding dimension $d_e$, the total number of training rounds N, and the sensitivity to the sparsity of the knowledge graph on the Dia dataset. We established the dimensions for entity embedding as $d_e, d_\tau \in \{100, 200, 300, 400, 500, 600, 700\}$ and the number of training rounds as $N \in \{100, 200, 300, 400, 500, 600\}$. The experimental parameters were harmonized with those previously described in the baseline and parameter configurations, except for the study-specific variables. The results of the experiment are visually depicted in Fig. 2.

Figure 2A shows the variation in the Hits@10 metric for our Tuck-KGC model as the entity vector dimensionality is altered. It is observed that with the increase in the entity embedding dimension, each model exhibits varying degrees of enhancement. The optimal performance is achieved at an entity embedding dimension of 400. Beyond this threshold, the model's efficacy tends to decline and stabilize. Our findings indicate that an entity embedding dimension of 400 yields the most favorable results, highlighting the critical role of this parameter in the model's performance. An insufficiently low dimension may result in underfitting, impeding the model's ability to effectively capture entity information,

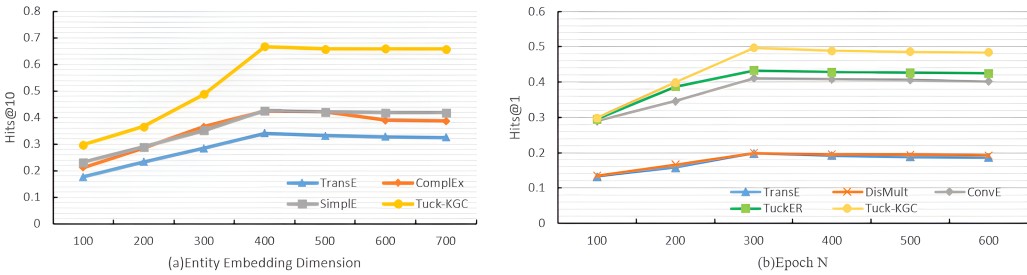

**Figure 2** **The variation trend for Tuck-KGC model as entity vector dimensionality and training iterations increase.**

while an excessively high dimension can lead to overfitting, thus compromising the model's performance.

Figure 2B shows the variation trend of the Hits@1indicator for our Tuck-KGC model as the training iterations increase. It is evident from the illustration that with the progression of training iterations, each model exhibits varying degrees of enhancement, culminating at the optimal performance when the iteration count reaches 300. Subsequently, surpassing this threshold leads to a gradual decline and stabilization of the model's outcomes. The findings suggest that the model achieves peak efficacy at 300 training iterations, highlighting the critical role of iteration count in influencing model performance. An insufficient number of iterations may result in underfitting, impeding the model's ability to effectively assimilate entity information, while an excessive number can lead to overfitting, detrimental to model performance.

To assess the sensitivity of Tuck-KGC to the sparsity of the knowledge graph, we randomly ignore triples from the Dia dataset and evaluate the effect on the entire test set. Figure 3 shows the test results of Tuck-KGC and the basic model on Hits@3. As depicted in Fig. 3, as the ratio of omitted triples increases, the performance of all models experiences a decline to varying extents. However, our method outperforms other baseline models throughout this decline, indicating a greater robustness to sparsity in our model compared to the baseline.

## Visualization of clustering entity type representations

We cluster the type embeddings using Kmeans and further implement dimensionality reduction using t-distributed scholastic neighbor embedding (t-SNE) for 2D visualization. As shown in Fig. 4A, different types of entities are clustered into separate categories in TuckER, while some clusters are close to each other because these entities share many common types. Figure 4B shows the clustering of entity embeddings of Tuck-KGC. It can be clearly observed that entity clustering with fused entity types can better distinguish entities, indicating that entity type embeddings can reflect the characteristics of entities. In other words, when an entity represents different meanings in different triples, it is difficult to distinguish. However, when given different types, the same entity has different entity type embeddings when encoded, so it is distinguishable. For example, we introduced a specific case in the "Introduction", the hypoglycemia entity type in the triple hypoglycemia,

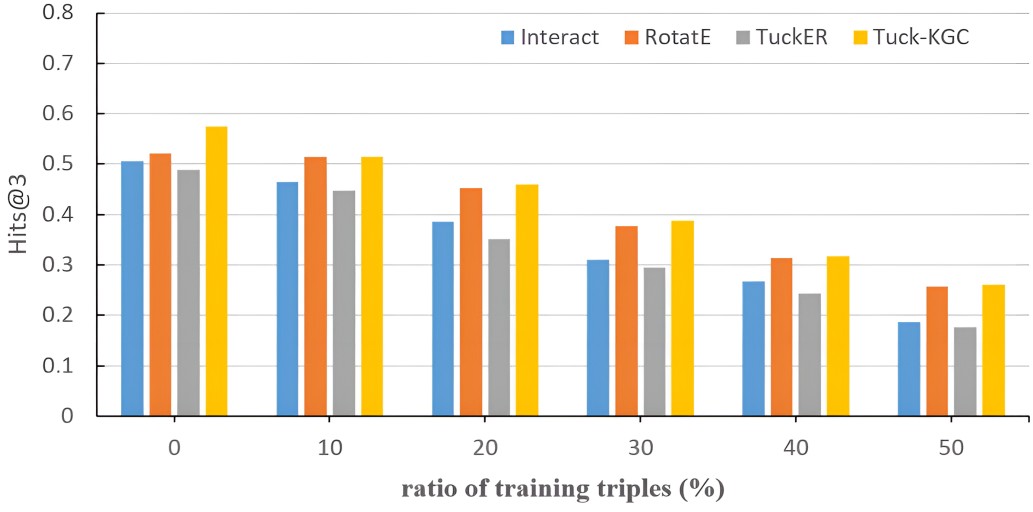

**Figure 3** The comparison results of Tuck-KGC and the basic model on the sensitivity of knowledge graph sparsity on Hits@3.

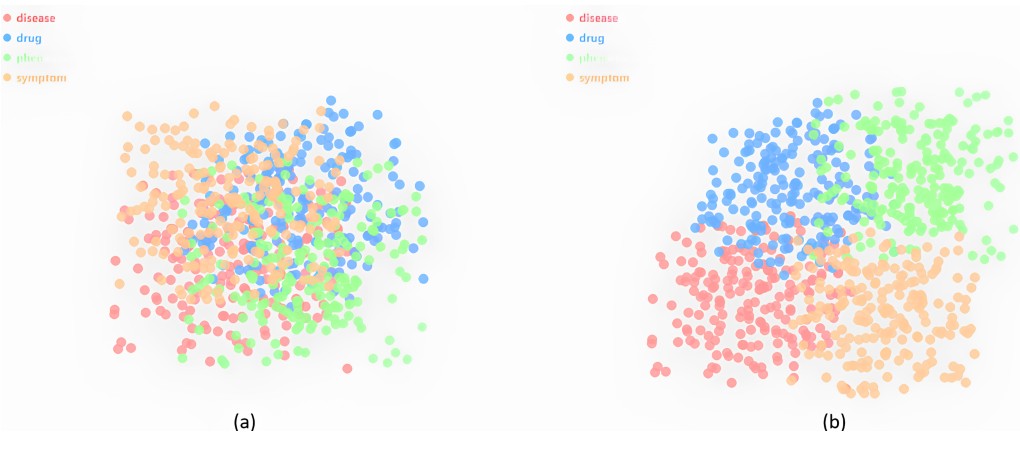

(a)                                  (b)

**Figure 4** The visualization of type embeddings clustering on Dia.

ADE_Disease, renal failure is a symptom of the disease, but hypoglycemia entity type in the triple fasting, Reason_Disease, hypoglycemia is the disease. These visualizations explain the effectiveness of using learned entity type embeddings in our knowledge graph.

## DISCUSSION

In the present study, we have constructed a knowledge graph encompassing 10,294 entities and 214 relationships pertaining to the diagnosis and treatment of diabetes through a combination of Chinese and Western medicine. Additionally, we introduce the Tuck-KGC knowledge graph complementation model, which incorporates entity types based on tensor decomposition. The inclusion of entity type embedding matrices in the model ensures the reasonableness and plausibility of the discriminative ternary relationships during training.

Tuck-KGC has demonstrated significant improvements across all evaluation metrics compared to knowledge graph complementation models such as TransE, DistMult, and SimplE, thereby demonstrating the efficacy of the Tuck-KGC model.

## SUGGESTION

In the present investigation, we confined our analysis to examining the influence of entity categories on the efficacy of the knowledge graph completion model, while other facets of the knowledge graph were not taken into account. Moving forward, our research efforts will be directed towards assessing the influence of other attributes of entities and relationships on the model's performance. Due to the limitations of the current data resources, the diabetes knowledge graph we have constructed is of modest scale, which hindered a comprehensive evaluation of the impact of incorporating the entity types on the model parameters. In the future, we plan to incorporate more knowledge about diabetes and its complications, focus on understanding how the size of the knowledge graph affects the model's efficiency, and aim to improve the efficiency of the model.

## ACKNOWLEDGEMENTS

The authors are deeply grateful to all those who contributed to this article and those who played a big role in the success of this article.

### Funding
This work was supported by 2023 Hunan Traditional Chinese Medicine Scientific Research Project (No. A2023048). The funders had no role in study design, data collection and analysis, decision to publish, or preparation of the manuscript.

### Grant Disclosures
The following grant information was disclosed by the authors:
2023 Hunan Traditional Chinese Medicine Scientific Research Project: No. A2023048.

### Competing Interests
The authors declare there are no competing interests.

### Author Contributions
- Jiangtao ZhangSun conceived and designed the experiments, performed the experiments, performed the computation work, prepared figures and/or tables, and approved the final draft.
- Yu Xin Yang analyzed the data, prepared figures and/or tables, and approved the final draft.
- Beiji Zou analyzed the data, authored or reviewed drafts of the article, and approved the final draft.

- Qinghua Peng analyzed the data, authored or reviewed drafts of the article, and approved the final draft.
- Xiao Xia Xiao conceived and designed the experiments, prepared figures and/or tables, authored or reviewed drafts of the article, and approved the final draft.

## Data Availability

The data are available at GitHub and Zenodo:

- https://github.com/zs1115/Tuck-KGC

- zs1115. (2024). zs1115/Tuck-KGC: 1.0 (1.0). Zenodo. https://doi.org/10.5281/zenodo.13169881.

## Supplemental Information

Supplemental information for this article can be found online at http://dx.doi.org/10.7717/peerj-cs.2522#supplemental-information.

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
