# Peer review of "Tuck-KGC: based on tensor decomposition for diabetes knowledge graph completion model integrating Chinese and Western medicine"

_PeerJ Computer Science, doi:10.7717/peerj-cs.2522_

## Round 0.1 · original submission · Major Revisions

Dear Authors

Your paper has been revised. Based on the reviewers' reports, major revisions are needed before it is considered for publication in PEERJ Computer Science.
The major issues are listed in what follows:
1) To improve the readability of your manuscript, the English language must be improved.
2) Authors should introduce the proposed Tuck-KCG method and the main contributions of this article in the abstract section. Furthermore, authors must clearly indicate the differences between the Tuck-KGG and the T-Tuck model to avoid confusion.
3) The summary of the existing methods must be widened.
4) The figures' titles are confusing, such as "Tuck-KGC for different entity embeddings size." The authors should add ordinate text to the line charts.
5) The titles of the tables in the article should be shorter, and some content can be written in the body paragraphs.
6) The parameter that produces the best result in terms of the model's validity performance must be clearly defined and reported in an appropriate table.

I advise the authors to consider all the reviewers' comments before resubmitting their revised paper.

Reviewer 1 ·

Basic reporting

- I suggest to proofread the paper.

- While the abstract is well-written, the introduction could benefit from some revisions. We suggest organizing the introduction in the following manner: First, discuss the limitations of the current methods, followed by an exploration of the potential advantages of employing Knowledge Graph Completion Model in this context. Then, delve into the applications of Knowledge Graph Completion Model in the domain of healthcare. This reorganization would strengthen the introduction and enhance its readability.
- Please no need to insert a formula in the Related work section.
- Please present all the hyperparmeters in a table.
- I think the manuscript can be further improved in terms of structure, content, and overall quality.

Experimental design

Implementation Codes Sharing: It is recommended to include the implementation codes of the project to facilitate the verification of the correctness of the implementation. Sharing the implementation codes would add transparency and credibility to the study (we cannot access to the provided links).

Validity of the findings

Please provide a viziualisation grpah of the results and interpret the relashionship.
Please provide valuable insights for your investigation

Additional comments

The manuscript lacks a section dedicated to suggestions for future research directions. Incorporating such a section titled "Suggestions for the Future" would enhance the manuscript's quality and provide valuable insights for further investigation and development in this area.

Reviewer 2 ·

Basic reporting

In the paper, the authors proposed diabetes knowledge graph completion model integrating Chinese and Western medicine based on tensor decomposition. However, there are still some questions needed to be addressed:
1. The English language should be improved. Such as the second sentence of the abstract and the first sentence of the introduction – the current phrasing makes comprehension difficult.
2. The fifth sentence of the abstract is not appropriate. The Dia dataset presented by the author cannot directly solve the problems of knowledge graph completion.
3. Authors should introduce the proposed Tuck-KCG method and the main contributions of this article in the abstract.
4. What is the T-Tuck model in the last sentence of abstract? The author needs to introduce briefly in abstract to avoid causing confusion.
5. The logic of the introduction section is confusing, and there is not enough summary of the existing methods.
6. The author should briefly describe each comparison method.
7. The titles of figures are confused, such as, “Tuck-KGC for different entity embeddings size”. And the author should add ordinate text to the line charts.
8. The titles of the tables in the article are too long, and some content can be written into the body paragraphs.

Experimental design

NA

Validity of the findings

NA

Reviewer 3 ·

Basic reporting

The paper discusses the knowledge graphs link prediction model for medicine. Specifically, the author presents a new dataset, which fused traditional Chinese and Western medicine (CWM dataset for short), and the Tuck-KGC model for link prediction on this diabetes knowledge graph. Overall, the manuscript is well structured, making the dataset preparation and the link prediction progress easy to capture the main idea. The introduction and related works are sufficient for understanding the context and motivation, but the author failed to define what is the entity information type. Additionally, the authors should carefully define the symbols and abbreviations used throughout the text. Additionally, the figure captions are inadequate and all of the figures should enhance the quality. The major limitation of the manuscript is the lack of attached raw data, and the provided dataset link is not accessible. Here is some mistakes that the paper should be correct:
- I suggest that you improve the line 23 for the “tensor decomposition based knowledge graph completion technology” because the explanation of methodology is only true for the link prediction task. Moreover, what does the term “knowledge graph complementation problem” mean? (Line 291)
- Next, I am confused about the name of your model, which could be Tuck-KGC (Line 32, 192, 355, …) or T-Tuck (36, 202, 207, …).
- On line 43 and 190, your definition about subject, object and relation in variable, vector or matrix should be clarified and well-formatted for better understanding.
- The model’s of DistMult (Yang et al, 2014) are incorrect on the line 137.
- Does Figure 1 illustrate the link prediction progress or the model architecture?.
- The Eq.4 on the line 215 shows the loss function that is incorrect for the $n_e$ variable (the $n$ value in the equation is used for the number of samples).
- There is missing Algorithm 1 definition (Line 277) and no mention for Figure 4.
- The Hits@K definition on the section Evaluation indicator has K in (1, 3, 10) not (1, 2, 3).
- The author missed to raise the importance of entity type information on what respect has the result been improved?

Experimental design

Overall, the author has clarified the research question and the literature review is sufficient to have the background on the knowledge graph, but there are some related works about considering the side information such as LiteralE. The implementation section does not mention the embedding size of entity and entity type but the experiment section has mentioned including the setup phase and results, it should be separated, while the author should be explicit why they need to take it into account.

Next, the paper does not mention the parameter for the best result in the case of the validity performance of the model. Furthermore, the position of the Link prediction subsection in the result section is inappropriate, this section should be present in front of the method section, and I recommend separating the manuscript into the background section with the knowledge graph formulation, link prediction, and knowledge graph embedding to be well-structured.

Validity of the findings

The paper investigates the gap of knowledge graph on medicine analysis specifically for diabetes, which lays the foundation for further contributions of the community in this field. Although the dataset statistics and entity types are mentioned sufficiently, the author needs to mention the entity types' characteristics such as imbalances, and distribution for each type and the self-constructed dataset should be written into steps for easy tracking. Moreover, the dataset attachment link is inaccessible, so we could not assess the real contributions and the quality of the dataset for this paper.

Finally, the conclusion demonstrates the importance of investigating dataset expansion but the suggested limitations of the author on knowledge expansion are not theory-supported, it should be strongly investigated in the experiment to validate the model scalability on this problem.

---

## Round 0.2 · Minor Revisions

Dear Authors,

Your paper has been reviewed. Some minor revisions are needed before it is considered for publication in PEERJ Computer Science. The comments of the reviewer who evaluated your manuscript are included in this letter. In particular, because the metrics used in your study do not explicitly show how the entity type embedding contributes to the overall performance, additional experiments are needed to clarify this point. I ask that you make changes to your manuscript based on those comments before uploading the revised manuscript.

Reviewer 4 ·

Basic reporting

1. The paper is clear, unambiguous, and written in professional English, making it easy to understand the authors' intentions and findings

2. The authors provide sufficient background and context for understanding their model. However, some operator definitions are omitted, which could hinder comprehension for some readers.

3. The article is well-structured and includes relevant figures and tables. However, some figure captions are missing.

4. The authors have shared the dataset and the PyTorch implementation of their model.

5. While some definitions are provided, others are missing or defined but not used, which could confuse readers looking for a comprehensive understanding of the theoretical framework.

Experimental design

The creation of the DIA dataset is well motivated and the efficiency of the model is well-discussed with respect to the Hits@k and MRR metrics. However, these metrics do not explicitly show how the entity type embedding contributes to the overall performance. Additional experiments or concrete facts on how Tuck-KGC leverages entity types compared to TuckER would be beneficial to readers.

Validity of the findings

See additional comments.

Additional comments

** Comment.
In this paper, the authors extend the tensor decomposition model TuckER by incorporating entity type embeddings. They achieve this by adapting the scoring function of TuckER. They evaluated their model, Tuck-KGC, on DIA, a new knowledge graph dataset that the authors created from the Western (DiaKG and SDKG-11) and Traditional Chinese Medicine diabetes datasets. This synergy has the potential to provide comprehensive diagnosis, treatment planning, personalised patient care and other unexpected benefits. Tuck-KGC shows good performance compared to baseline models such as TransE, RotatE and TuckER.

** Question

1. In the Tuck-KGC Implementation section, the authors define the A, B, C and Gamma matrices, but these matrices are not used anywhere in the paper. If these variables are not needed, I would recommend that the authors remove them.

2. Figures 2 and 3 have no captions.

3. In equation 2, the authors have used operators that are not defined. I would suggest that the authors add these definitions to the paper.
Still in equation 2, it can be observed that the embeddings of the head and tail entity types are both called by e_tau. Does this mean that the authors have assumed that entities that occur in the same triples are of the same type? If not, the authors should make this clear in the text.

4. In Table 4 we can see that Tuck-KGC significantly outperforms RotatE on the evaluation metrics Hits@1, Hits@3 and Hits@10. Surprisingly, the MRR of Tuck-KGC is relatively lower than the MRR of RotatE. I would like the authors to explain how this is possible?

5. Although the authors consider a considerable number of basic models, none of them are designed to learn from entity type. Meanwhile, there are models such as TransET (briefly described by the authors in the Methods/Knowledge Embedding section, but incorrectly named TransE) and AutoETER (arXiv preprint arXiv:2009.12030 published in 2020) that are more suitable for benchmarking and might be better candidates to complete DIA. It is crucial to include some of these models.

---

## Round 0.3 · Minor Revisions

Dear Authors,
Your paper has been revised. Minor revisions are needed before it is considered for publication in PEERJ Computer Science. Please address the reviewers' concerns and resubmit your article once you have updated it. In particular, you must clarify the meaning of some statements (i.e., "types of entity embeddings are different") to improve the readability of your work.

Reviewer 4 ·

Basic reporting

Not Applicable

Experimental design

Not Applicable

Validity of the findings

Not Applicable

Additional comments

I read the authors' response to my reviews with interest. I observed updates to the GitHub page, yet I haven't verified the functionality of the framework by running the code. Additionally, the authors introduced a section titled 'Visualization of Clustering Entity Type Representations,' highlighting the significance of entity type embeddings. However, I was a bit confused when I read the sentence "the same entity but different types of entity embeddings are different" since I can see from the DIA dataset that each entity has exactly one entity type. Could the authors clarify what is meant by "types of entity embeddings are different"?

I recommend refining this section for clarity.

---

## Round 0.4 · accepted · Accept

Dear Authors,

Your paper has been accepted for publication in PeerJ Computer Science.

Thank you for your fine contribution.